# THE CHALLENGES OF EXPLORATION FOR OFFLINE REINFORCEMENT LEARNING

## ABSTRACT

Offline Reinforcement Learning (ORL) enables us to separately study the two interlinked processes of reinforcement learning: collecting informative experience and inferring optimal behaviour. The second step has been widely studied in the offline setting, but just as critical to data-efficient RL is the collection of informative data. The task-agnostic setting for data collection, where the task is not known *a priori*, is of particular interest due to the possibility of collecting a single dataset and using it to solve several downstream tasks as they arise. We investigate this setting via curiosity-based intrinsic motivation, a family of exploration methods which encourage the agent to explore those states or transitions it has not yet learned to model. With Explore2Offline, we propose to evaluate the quality of collected data by transferring the collected data and inferring policies with reward relabelling and standard offline RL algorithms. We evaluate a wide variety of data collection strategies, including a new exploration agent, Intrinsic Model Predictive Control (IMPC), using this scheme and demonstrate their performance on various tasks. We use this decoupled framework to strengthen intuitions about exploration and the data prerequisites for effective offline RL.

## 1 INTRODUCTION

The field of offline reinforcement learning (ORL) is growing quickly, motivated by its promise to use previously-collected datasets to produce new high-quality policies. It enables the disentangling of collection and inference processes underlying effective RL (Riedmiller et al., 2021). To date, the majority of research in the offline RL setting has focused on the inference side - the extraction of a performant policy given a dataset, but just as crucial is the development of the dataset itself. While challenges of the inference step are increasingly well investigated (Levine et al., 2020; Agarwal et al., 2020), we instead investigate the collection step. For evaluation, we investigate correlations between the properties of collected data and final performance, how much data is necessary, and the impact of different collection strategies. Whereas most existing benchmarks for ORL (Fu et al., 2020; Gulcehre et al., 2020) focus on the single-task setting with the task known *a priori*, we evaluate the potential of task-agnostic exploration methods to collect datasets for previously-unknown tasks. Task-agnostic data is an exciting avenue to pursue to illuminate potential tasks of interest in a space via unsupervised learning. In this setting, we transfer information from the unsupervised pretraining phase not via the policy (Yarats et al., 2021) but via the collected data.

Historically the question of how to act - and therefore collect data - in RL has been studied through the exploration-exploitation trade-off, which amounts to a balance of an agent's goals in solving a task immediately versus collecting data to perform better in the future. Task-agnostic exploration expands this well-studied direction towards how to explore in the absence of knowledge about current or future agent goals (Dasagi et al., 2019). In this work, we particularly focus on intrinsic motivation (Oudeyer & Kaplan, 2009), which explores novel states based on rewards derived from the agent's internal information. These intrinsic rewards can take many forms, such as curiosity-based methods that learning a world model (Burda et al., 2018b; Pathak et al., 2017; Shyam et al., 2019), data-based methods that optimize statistical properties of the agent's experience (Yarats et al., 2021), or competence-based metrics that extract skills (Eysenbach et al., 2018). In particular, we perform a wide study of data collected via curiosity-based exploration methods, similar to ExORL (Yarats et al., 2022). In addition, we introduce a novel method for effectively combining curiosity-based rewards with model predictive control.

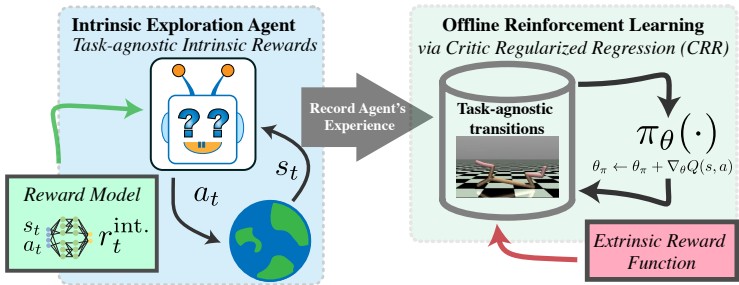

Figure 1: The Explore2Offline framework for evaluating data-efficient intrinsic agents. First the agent acts in the environment task-agnostically to search for novel states. After a set lifetime, the agent experience stored in a replay buffer is labeled with the rewards of a task of interest. This replay buffer is used to train an RL policy with the offline reinforcement learning algorithm Critic Regularized Regression in order to finally evaluate the quality of exploration in the environment.

In Explore2Offline, we use offline RL as a mechanism for evaluating exploration performance of these curiosity-based models, which separates the fundamental feedback loop key to RL in order to disentangle questions of collection and inference Riedmiller et al. (2021) as displayed in Fig. 1. With this methodology, our paper has a series of contributions for understanding properties and applications of data collected by curiosity-based agents.

**Contribution 1**: We propose Explore2Offline to combine offline RL and reward relabelling for transferring information gained in the data from task-agnostic exploration to downstream tasks. Our results showcase how experiences from intrinsic exploration can solve many tasks, partially reaching similar performance to state-of-the-art online RL data collection.

**Contribution 2**: We propose Intrinsic Model Predictive Control (IMPC) which combines a learned dynamics model and a curiosity approach to enable online planning for exploration to minimize the potential of stale intrinsic rewards. A large sweep over existing and new methods shows where task-agnostic exploration succeeds and where it fails.

**Contribution 3**: By investigating multi-task downstream learning, we highlight a further strength of task-agnostic data collection where each datapoint can be assigned multiple rewards in hindsight.

## 2 RELATED WORKS

### 2.1 CURIOSITY-DRIVEN EXPLORATION

Intrinsic exploration is a well studied direction in reinforcement learning with the goal of enabling agents to generate compelling behavior in any environment by having an internal reward representation. Curiosity-driven learning uses learned models to reward agents that reach states with high modelling error or uncertainty. Many recent works use the prediction error of a learned neural network model to reward agents' that see new states Burda et al. (2018b); Pathak et al. (2017). Often, the intrinsic curiosity agents are trained with on-policy RL algorithms such as Proximal Policy Optimization (PPO) to maintain recent reward labels for visited states. Burda et al. (2018a) did a wide study on different intrinsic reward models, focusing on pixel-based learning. Instead, we use off-policy learning and re-label the intrinsic rewards associated with a tuple when learning the policy. Other strategies for using learned dynamics models to explore is to reward agents based on the variance of the predictions Pathak et al.; Sekar et al. or the value function (Lowrey et al., 2018). We build on recent advancements in intrinsic curiosity with the Intrinsic Model Predictive Control agent that has two new properties: online planning of states to explore and using a separate reward model from the dynamics model used for control.

### 2.2 UNSUPERVISED PRETRAINING IN RL

Recent works have proposed a two-phase RL setting consisting of a long "pretraining" phase in a version of the environment without rewards, and a sample-limited "task learning" phase with visible rewards (Schwarzer et al., 2021). In this setting the agent attempts to learn task-agnostic information about the environment in the first phase, then rapidly re-explore to find rewards and produce a policy

specialized to the task. Various methods have addressed this setting with diverse policy ensembles, such as a policy conditioned on a random variable whose marginal state distribution exhibits high coverage (Eysenbach et al., 2018) or finding a set of policies with diverse successor features (Hansen et al., 2020). Similarly Liu & Abbeel (2021) learn a single policy which approximately maximizes the estimated entropy of its state distribution in a contrastive representation space. Another strategy collects diverse data during the pretraining phase and uses it to learn representations and exploration rewards that are beneficial for downstream tasks (Yarats et al., 2021). A central focus of all of these methods is delivering agents which can explore efficiently at task learning time. While the pretraining and data collection phases of unsupervised RL pretraining and Explore2Offline (respectively) are similar, in Explore2Offline the task learning phase is performed on relabeled offline transitions, enabling us to use information acquired throughout training and not only the final policy.

## 2.3 EVALUATING TASK-AGNOSTIC EXPLORATION

While there are many proposed methods for exploration, evaluation of exploration methods is varied. Recent work in exploration have proposed a variety of evaluation metrics, including fine-tuning of agents post-exploration (Laskin et al., 2021), sample-efficiency and peak performance of online RL (Whitney et al., 2021), zero-shot transfer of learned dynamics models (Sekar et al.), multi-environment transfer (Parisi et al., 2021), and skill extraction to a separate curriculum (Groth et al., 2021). Task-agnostic exploration has been investigated via random data (Cabi et al., 2019) and intrinsic motivation as a source of data for offline RL (Dasagi et al., 2019; Endrawis et al., 2021), but has only been evaluated in the single-task setting and limited by current ORL implementations. Offline RL is a compelling candidate for evaluating exploration data because of its emerging ability to generalize across experiences in addition to imitating useful behaviors.

**Complementary work** echoes the importance of data collection for offline RL from the perspective of unsupervised RL (Yarats et al., 2022), while our work focuses more on the relationship between the exploration challenges of an environment and how a new exploration algorithm could address current data generation shortcomings.

## 2.4 OFFLINE REINFORCEMENT LEARNING

With Offline Reinforcement Learning, we decouple the learning mechanism from exploration by training agents from fixed datasets. Various recent methods have demonstrated strong performance in the offline setting (Wang et al., 2020; Kumar et al., 2020; Peng et al., 2019; Fujimoto & Gu, 2021). In Explore2Offline we use a variant of Critic Regularised Regression (Wang et al., 2020).

Many datasets and benchmarks such as D4RL (Fu et al., 2020) and RL Unplugged (Gulcehre et al., 2020) have been proposed to investigate different approaches. The use of offline datasets has even been extended to improve online RL performance (Nair et al., 2020). Our goal is related; instead of investigating multiple offline RL approaches, we investigate mechanisms to generate datasets for downstream tasks. Analysis over the desired state-action and reward distributions for ORL are studied, but little work is done to address how best to generate this data (Schweighofer et al., 2021).

On the theory side, recent works have investigated the Explore2Offline setting, which they call "reward-free exploration" (Jin et al., 2020; Kaufmann et al., 2021). These works study algorithms which guarantee the discovery of $\varepsilon$-optimal policies after polynomially many episodes of task-agnostic data collection, though the algorithms they study are not straightforwardly applicable to the high-dimensional deep RL setting with function approximation.

## 2.5 REWARD RELABELLING

By using off-policy or even offline learning, data generated for one task and reward can be applied to learn a variety of potential tasks. In off-policy RL, we can identify useful rewards for an existing trajectory based on later states from the same trajectories (Andrychowicz et al., 2017), uncertainty over a trajectory (Nasiriany et al., 2021), distribution of goals (Nasiriany et al., 2021), related tasks (Riedmiller et al., 2021; Wulfmeier et al., 2019), agent-intrinsic tasks (Wulfmeier et al., 2021), via inverse reinforcement learning (Eysenbach et al., 2020) as well as other mechanisms (Li et al., 2020).

In the context of pure offline RL, we can go one step further as we are not required to find the optimal tasks for stored trajectory data. In this setting, data can be used for learning with a massive set of rewards such as all states visited along stored trajectories (Chebotar et al., 2021). We will evaluate our approaches for exploration across downstream tasks and relabel data with all possible tasks.

## 3 METHODOLOGY

### 3.1 REINFORCEMENT LEARNING

Reinforcement Learning (RL) is a framework where an agent interacts with an environment to solve a task by trial and error. The objective of an agent is often to maximize the cumulative future reward on a predetermined task, $\mathbb{E}\left[\sum_{\tau=0}^{\infty} \gamma^{\tau} r_{\tau} | s_0 = s_t\right]$. We utilize the setting where an agent's interactions with an environment are modeled as a Markov Decision Process (MDP). A MDP is defined by a state of the environment $s$, an action $a$ that is taken by an agent according to a policy $\pi_\theta(s_t)$, a transition function $p(s_{t+1}|s_t, a_t)$ governing the next state distribution, and a discount factor $\gamma \in [0, 1]$ weighting future rewards. With a transition in dynamics, the agent receives a reward $r_t$ from the environment and stores the SARS data in a dataset $\mathcal{D} : \{s_k, a_k, r_k, s_{k+1}\}$. Alternatively to this environment-centric reward formulation is the concept of intrinsic rewards, where the agent maximizes an internal notion of reward in an task-agnostic manner to collect data.

### 3.2 CURIOSITY-DRIVEN EXPLORATION

**Existing Methods**  Reaching new, valuable areas of the state-space is crucial to solving sparse tasks with RL. One method to balance attaining new experiences, exploration, with the goal of solving a task, exploitation, is using curiosity models. Curiosity models are a subset of intrinsic rewards an agent can use to explore by creating a reward signal, $r^{\text{int.}}$. These models encourage exploration by optimizing the signal from a learned model that corresponds to a modeling error or uncertainty, which often occurs at states that have not been visited frequently. We deploy a series of intrinsic models: the simplest, Next Step Model Error maximizes the error of a learned one-step model $r^{\text{int.}} = \|\hat{s}_{t+1} - s_{t+1}\|^2$, Random Network Distillation (RND) maximizes the distance of a learned state encoding to that of a static encoding $r^{\text{int.}} = \|\hat{\eta}(s_t) - \eta(s)\|^2$ (Burda et al., 2018b), the Intrinsic Curiosity Module (ICM) maximizes the error on a forward dynamics model learned in the latent space, $\phi(s)$, of a inverse dynamics model $r^{\text{int.}} = \|\hat{\phi}_t - \phi\|^2$ (Pathak et al., 2017), and Dynamics Disagreement (DD) maximizes the variance of an ensemble of learned one-step dynamics models $r^{\text{int.}} = \sigma(\hat{s}_{t+1}^i)$ (Pathak et al.).

**Intrinsic Model Predictive Control**  Model Predictive Control (MPC) on a learned model has been used for control across a variety of simulated an real world settings (Wieber; Camacho & Alba, 2013), including recently with model-based reinforcement learning (MBRL) algorithms (Williams et al., 2017; Chua et al., 2018; Lambert et al., 2019). MBRL using MPC is an iterative loop of learning a predictive model of environment dynamics $f_\theta(\cdot)$ (e.g. a one-step transition model), and acting in the environment through the use of model based planning with the learned

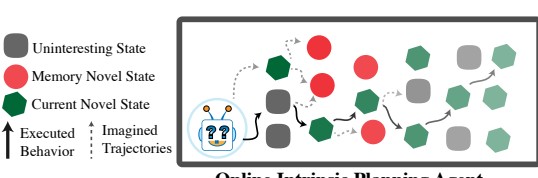

**Online Intrinsic Planning Agent**

Figure 2: A conceptual depiction of an exploration with foresight via planning. By simulating the future with a learned dynamics model, ideally an agent should be able to avoid damaging states before having experienced them to collect interesting data in a sample-efficient manner.

model. This planning step usually involves optimizing for a sequence of actions that maximizes the expected future reward (Eqn. 1), for example, via sample based optimization; the MPC loop executes the first action of this sequence followed by replanning.

$$a = \arg\max_{a_{t:t+\tau}} \sum_t^\tau r(\hat{s}_t, a_t), \quad s.t. \quad \hat{s}_{t+1} = f_\theta(s_t, a_t). \tag{1}$$

| Curiosity Model | Training Labels | Eval. Labels | MPC Compatible |
|---|:---:|:---:|:---:|
| Random Network Distillation (RND) | $\{s\}$ | $\{s\}$ | **Yes** |
| Intrinsic Curiosity Module (ICM) | $\{s, a, s'\}$ | $\{s, a, s'\}$ | No |
| Next Step Model Error (NSM) | $\{s, a, s'\}$ | $\{s, a, s'\}$ | No |
| Dynamics Dissimilarity (DD) | $\{s, a, s'\}$ | $\{s, a\}$ | **Yes** |

Table 1: The intrinsic models used with the components of a tuple needed for training and evaluation. To function with planning, an intrinsic model cannot need access to the next state to compute reward.

The reward function $r$ defines the behavior of the planned sequence of actions in model-based planning. For task-specific RL this can be the task reward function (known or estimated from data). Instead, our Intrinsic MPC (IMPC) agent uses a curiosity based reward for planning in order to encourage task agnostic exploration by reasoning about what states are currently interesting and novel. The goal of planning being used to visit new interesting states, rather than states that were recorded with high intrinsic reward in the replay memory is shown in Fig. 2. This evaluation occurs by sampling action sequences, unrolling them using the forward dynamics model, scoring the rollouts with the learned curiosity model, and finally taking the first action of the sequence with the highest score. Given that this evaluation happens with access to only imagined states and a proposed action, only a subset of intrinsic models can be used with planning, as summarized in Tab. 1. We primarily evaluate IMPC using the RND curiosity model, but we also present results with the DD model.

We use the Cross Entropy Method (CEM) (De Boer et al.), a sample based optimization procedure for planning. Inspired by prior work Byravan et al. (2021) we use a policy to generate action candidates for the planner; this policy is trained using the Maximum a-posteriori Policy Optimization (MPO) algorithm (Abdolmaleki et al., 2018) from data generated by the MPC actor. Additionally, to amortize the cost of planning we interleave planning with directly executing actions sampled from the learned policy. This is achieved by specifying a planning probability $0 < \rho < 1$; at each step in the actor loop we choose either to plan or execute the policy action according to $\rho$ (we use $\rho = 0.9$). Additional algorithmic details are included in Appendix A.1.

### 3.3 OFFLINE REINFORCEMENT LEARNING

To train an agent offline from task-agnostic exploration data, we determine rewards from observations in hindsight. While the approach relies on the ability to compute rewards based on observations, a large set of tasks can be described in this manner (Li et al., 2020). In comparison to online learning, we have the benefit that we do not need to determine for which task data is most informative given a commensurate number of tasks. Instead we can relabel the data with all possible rewards to maximise its utility. Given the new task rewards, we replace the intrinsic reward in our trajectory data and apply a variant of a recent state-of-the-art offline RL algorithm, Critic Regularised Regression (CRR) Wang et al. (2020). While we apply CRR for our investigation, the overall method is general in that it could be applied with other approaches. We iteratively update critic and actor optimising their respective losses following Equation 2 and 3.

$$\mathcal{L}_Q = \mathbb{E}_{\mathcal{B}}\Big[D\big(Q_\theta(s_t, a_t), (r_t + \gamma \mathbb{E}_{a_t \sim \pi(s_{t+1})} Q_{\theta'}(s_{t+1}, a_t))\big)\Big], \tag{2}$$

Since we use a distributional categorical critic, we apply the divergence measure $D$ instead of the squared Euclidean loss, following (Bellemare et al.). With $f = \text{ReLU}(\hat{\mathcal{A}}(s_t, a_t))$ and $\hat{\mathcal{A}}$ the advantage estimator via $Q_\pi(s_t, a_t) - 1/m \sum_{i=1}^N Q_\pi(s_t, a_i)$, the policy is optimized as:

$$\pi(a_t|s_t) = \arg\max_\pi \mathbb{E}_{(s_t, a_t) \sim \mathcal{B}}\Big[f(Q_\pi, \pi, s_t, a_t) \log \pi(a|s)\Big]. \tag{3}$$

## 4 EXPERIMENTS

### 4.1 EXPERIMENTAL SETTING

**Exploration Agents** In this work we benchmark a variety of task-agnostic exploration agents. We classify agents as reactive, selecting actions with a policy, or planning, selecting actions by

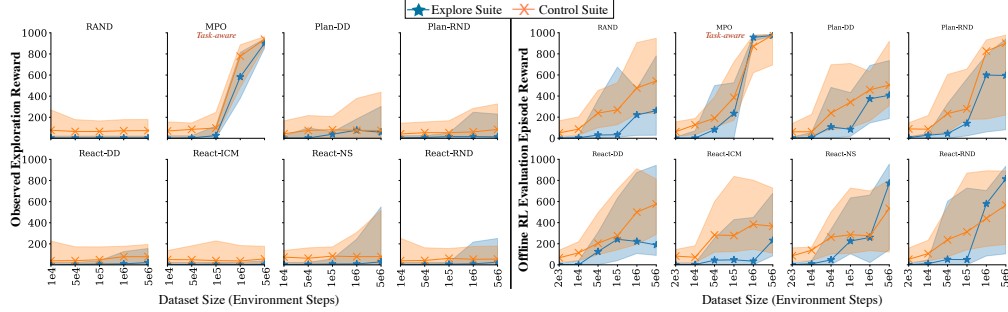

Figure 4: Comparing the collected rewards of an exploration agent (*left*) to the reward achieved when training an offline RL algorithm on the same data (*right*). There is a substantial gain in performance with offline RL algorithms simply by having more data, irregardless as to if the data has higher density of observed rewards. The median, 90[th] and 10[th] percentiles are shown for each agent, combined across tasks. *Note*: the MPO agent learns online and is the only task-aware method.

optimizing over trajectories. The reactive agents are trained with Maximum Apriori Optimization (MPO) (Abdolmaleki et al., 2018) and include the curiosity models RND, DD, ICM, and NS. We compare these to IMPC with DD and RND optimized with CEM and a fixed random agent that samples from the action distribution. Some figures will include a label of MPO, which corresponds to the benchmark of the *task-aware* RL agents, which provides interesting context to Explore2Offline. The online agent represents the state-of-the-art performance when the task is known a priori – matching it without an environment reward function would highlight the potential of Explore2Offline.

**Environments** In this work, we investigate the exploration performance of a variety of DeepMind Control Suite tasks (Tassa et al., 2018). We evaluate 4 domains (Ball-in-Cup, Finger, Reacher, and Walker) including 14 tasks with a variety of state-action sizes, with further details included in the Appendix. In order to include more challenging environments for task-agnostic exploration, we use modifications proposed by Whitney et al. (2021), Explore Suite, which include constrained initial states and sparser reward functions.

### 4.2 TASK-AGNOSTIC DATA COLLECTION

We compare the amount of task-reward received per-episode across a variety of agents and tasks, which is an intuitive metric for exploration agent performance but can only act as a proxy for exploration. To evaluate task-agnostic reward, the exploration agents are run without access to the environment rewards, with the rewards are relabelled after. The distributions of normalized reward achieved during the 5000 training episodes for four example tasks are documented for all of the exploration

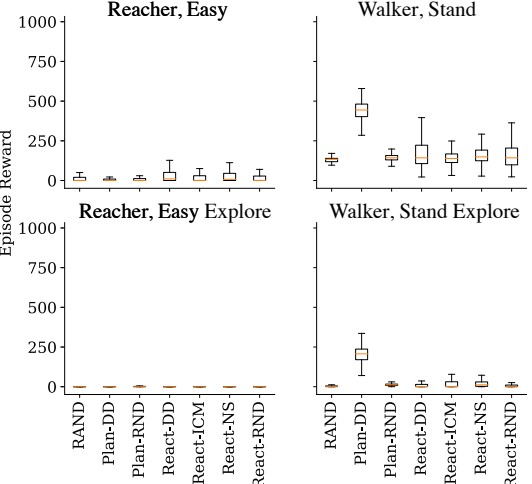

Figure 3: Collected reward per-episode (1000 env. steps) distributions across a subset of tasks and their Explore Suite variants. Boxplots show the median episode reward (red line), 25th and 75th percentiles (box), and the maximum and minimum reward computed over 5000 training episodes (whiskers). Only a horizontal line indicates that the episode reward is equal or near 0.

agents in Fig. 3. Crucially, the Explore Suite variants are challenging for the random agent across its lifetime, where all of the examples shown have median episode-reward of 0. There is a wide diversity in the agent-task pairings by proxy of measuring experienced reward, showcasing the large potential of future work to better understand this area.

To give an overall view of data collection for each agent, we show in Fig. 4 (*left*) the Control and Explore Suite average collected reward across all tasks. The median, 90[th] and 10[th] percentiles are

| Domain, Task | Best Task-Agnostic | Random Agent | IMPC-RND | Task-Aware |
|---|---|---|---|---|
| Ball-in-Cup, Catch | 973.28 | 943.04 | 973.40 | **984.11** |
| Ball-in-Cup, Catch Exp. | 962.56 | 0.00 | 938.59 | **974.27** |
| Finger, Turn Easy | **981.76** | 822.27 | **981.76** | **983.16** |
| Finger, Turn Easy Exp. | 947.94 | 915.97 | 944.22 | **968.67** |
| Finger, Turn hard | **975.33** | 543.57 | **975.33** | **977.27** |
| Finger, Turn hard Exp. | 952.63 | 690.79 | 595.06 | **965.60** |
| Reacher, Easy | **955.01** | **955.01** | 900.67 | 766.79 |
| Reacher, Easy Exp. | 813.91 | 612.77 | 740.70 | **973.91** |
| Reacher, Hard | **719.90** | 428.38 | 719.90 | 587.37 |
| Reacher, Hard Exp. | 930.95 | 261.20 | 410.78 | **973.91** |
| Walker, Stand | 534.99 | 297.63 | 302.12 | **991.73** |
| Walker, Stand Exp. | 258.55 | 243.53 | 81.28 | **982.79** |
| Walker, Walk | 445.53 | 146.76 | 96.88 | **978.36** |
| Walker, Walk Exp. | 426.07 | 47.54 | 78.88 | **975.06** |

Table 2: The best performance with $5 \times 10^6$ transition datasets for all 7 task-agnostic exploration agents, the specific performance of Random Agent and IMPC-RND (to illustrate strengths and weaknesses), and the task-aware MPO agent (the reported number is the median across 3 ORL seeds on a fixed dataset). Bold represents the best reward across a task and the median of near-best agents being within the $10^{th}$-to-$90^{th}$ percentile range of the best agent. This shows where exploration agents perform similarly and consistently to online RL for dataset collection. Even for some tasks where intrinsic agents are near to task-aware exploration, the online RL has much tighter performance across seeds. Results for all agents are shown in Appendix A.3.

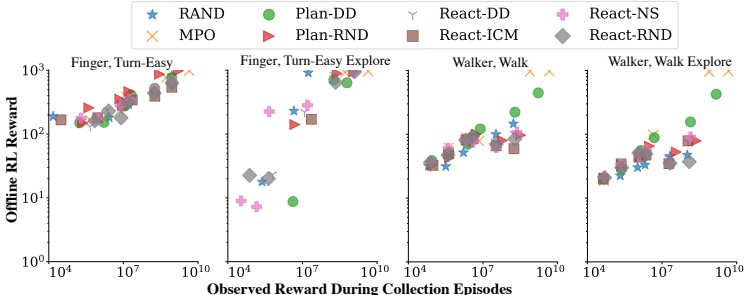

Figure 5: Correlation of final offline RL performance to cumulative reward in the training set on two example tasks and their explore-suite variants. The ORL performances are the median across 3 trials for each task and agent. As the dataset sizes we use span many orders of magnitude of samples, both axes are plotted on a log-scale. Across all tasks, there is a trend of more reward in the training distribution relating to a better performing CRR agent.

shown for each agent, combined across tasks. Here we also show what reward a task-aware agent (MPO) will collect in its lifetime. It is included to indicate an upper target for exploration, rather than a competitive baseline. An important artifact here is the random agent's flatness of reward achieved across time – the other exploration agents show an increase in median reward as the dataset size grows (especially on Explore Suite). This change in reward distribution across dataset size indicates a diversity of behaviors in the intrinsic exploration agents, while the random agent receives reward from the same distribution repeatedly. This can be seen as a curiosity-based agent such as IMPC with RND achieves varied rewards and the random agent has a repeated reward distribution.

While the collected reward can be a indicator of the usefulness of an exploration agent, it is not directly transferable to a task-focused policy capable of solving tasks. Without careful environment design, task-agnostic agents focusing on novel states will cover the entire state-space regardless of the predefined downstream task.

## 4.3 EVALUATING OFFLINE RL ON EXPLORATION DATA

We study the performance of intrinsic agents via a SOTA offline reinforcement learning algorithm, Critic Regularized Regression (CRR) (Wang et al., 2020). For each agent, 3 policies were trained on dataset sizes within the range of $2 \times 10^3$ to $5 \times 10^6$ environment steps. The mean of the performance across all explore and control suite tasks with confidence intervals is shown in Fig. 4 (*right*). Due to the width of this study, we were only able to evaluate the offline RL performance on 3 seeds per each collected dataset – the median, max and min across these three policies from CRR are shown in each subplot. Here, we see three core findings that we will continue to detail: 1) for dataset sizes < $1 \times 10^5$, there is little benefit to task-aware learning for data collection and the random agent is a strong baseline versus other intrinsic model-based agents; 2) on larger datasets, the task-aware RL method, MPO, jumps ahead of the explorers, but the exploration agents all continue to improve in offline RL performance with more data; 3) the novel IMPC approach with RND, along with existing methods of MPO with RND or DD, performs best on average with the largest datasets.

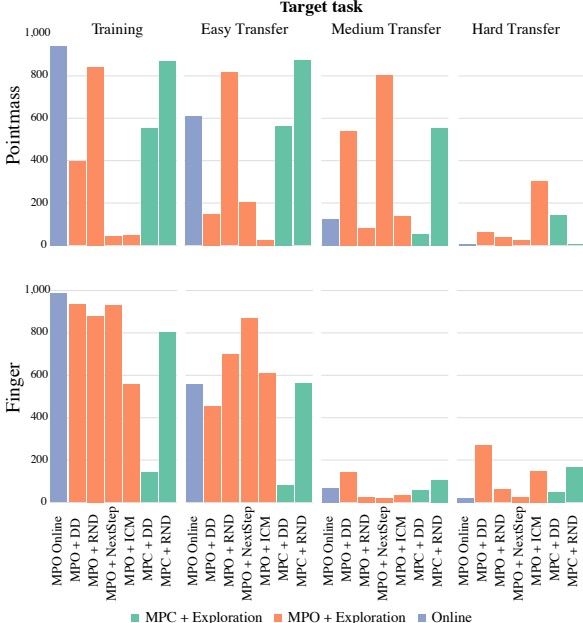

Figure 6: Performance across tasks in the Finger and Pointmass domains for offline RL agents trained on a single dataset. From left to right the tasks become qualitatively less similar to the initial task. Task-agnostic exploration regimes show the potential for better performance over task-aware learning, but the results are varied across task.

To showcase which tasks are solved by the Explore2Offline framework, we document the median performance of the exploration agents with the full $5 \times 10^6$ steps training set size versus the task-aware MPO in Tab. 2. Explore2Offline with these agents solve all but the Walker domain tasks, with further results included in the Appendix.

To highlight why dataset size and observed reward are such powerful indicators of ORL performance, we show in Fig. 5 the correlation for the Finger Turn and Walker Walk tasks of the cumulative reward in a dataset versus the offline RL performance for that dataset. There is a clear trend of more reward resulting in a better policy for the tasks paired with the environment. Fig. 7 uses Spearman's rank correlation to visualise how dataset size is a considerably better predictor of performance than any reward statistics including mean, sum or 80% quantile. This further emphasises the importance to transfer via increasingly large datasets instead instead of the final exploration policy. In the next section we will evaluate how re-using data from task-agnostic exploration can enable multi-task performance.

## 4.4 EVALUATING TASK-AGNOSTIC DATA FOR MULTITASK RL

A key motivation for collecting task-agnostic data is its applicability when there are a variety of downstream tasks of interest, including those which might not be known at data collection time. In the ideal case, task-agnostic exploration could collect a single dataset, then offline RL could consume that data (along with relabeled rewards) to solve arbitrary tasks. We evaluate the quality of datasets collected by various exploration agents for use with multiple downstream tasks.

For this evaluation we collected one dataset for the Pointmass and Finger environments using each of seven exploration agents, then trained policies for downstream tasks by relabeling the data with different reward functions. Each of the tasks is defined by a sparse +1 reward corresponding to a particular goal state. As a baseline, the Online agent collects experience using a standard online RL algorithm as it learns to solve a "Training" task, while all of the other data collection agents are task agnostic. The datasets collected by each agent are evaluated with offline RL on the "Training" task

and three others: "Easy Transfer", "Medium Transfer", and "Hard Transfer". These datasets are ranked in the level of challenge that a task-aware agent has in generalization. Tasks increase in challenge when goals require moving farther away from the training goal, which can be either travelling further in the same direction (often easier) or entirely misaligned (for harder generalization). Due to the dynamics of the systems, farther-away targets may be easier to discover, e.g. the "Medium Transfer" target for Pointmass lies in the corner of the arena. Full details of these environments, along with experiments on three more, are available in Appendix A.4.

The performance of the exploration agents and the task-aware MPO agent are varied across the tasks as shown in Fig. 6. Depicted is the mean reward achieved of offline RL policies trained on 3 random seeds for each of 3 datasets of $5 \times 10^6$ transitions. While collecting data specifically for the target downstream task is the best option when the data will be used only on that task, the task-aware performance can degrade on even a slightly misaligned test-task when compared to task-agnostic counterparts. The potential for multitask transfer of exploration agents is highlighted, but further work is needed in more open ended environments to show the potential of Explore2Offline.

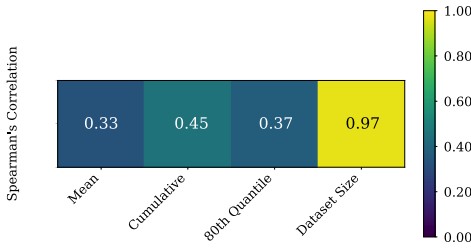

Figure 7: Spearman's rank correlation between performance of the ORL policy and data statistics such as reward (mean, cumulative and $80^{\text{th}}$ quantile) as well as dataset size for task-agnostic datasets.

## 5  DISCUSSION

Explore2Offline points to interesting directions for further understanding and utilizing task-agnostic exploration agents. To start, there are two trends that point to a need for further work on exploration methods. On average across our evaluation suite, the random agent performs very closely to the curiosity-based methods, and any particular exploration method varies substantially across task. The performance of the random agent suggests some similarities in the data collected by the random agent and the curiosity-based methods. As mentioned previously, curiosity-based methods are exhaustive (given enough time), and do not consider useful trends that may be common in downstream tasks. This shows a need for future exploration methods to be able to prioritize interesting subsets of a state-space and generalize across domains to create flexible agents.

Although our evaluation demonstrates the potential of using offline RL on task-agnostic data, there is substantial variation across task-agent pairings with the chosen static offline RL algorithm (CRR). This variation needs to be studied in more detail to better understand the limitations posed by the algorithm, and differentiate them from the quality of the data itself.

The intrinsic MPC agent can be progressed by utilizing it in other forms of deep RL evaluation. By transferring a learned dynamics model, this flexible exploration agent could also be evaluated as a task-aware agent (*i.e.* zero-shot learning of a new task) or in online RL by weighting the intrinsic reward model and the environment reward (*i.e.* better explore-exploit balance).

## 6  CONCLUSION

We introduce Explore2Offline, a method for utilizing task-agnostic data for policy learning of unknown downstream tasks. We describe how an agent can be used to collect the requisite data once for solving multiple tasks, and demonstrate performance comparable to an online learning agent. Additionally, we show policies trained on task-agnostic data may be robust to variations to the initial task compared to task-aware learning, resulting in better transfer performance. Finally, data from the new exploration agent, Intrinsic Model Predictive Control, performs strongly across many tasks. As offline RL emerges as a useful tool in more domains, a deeper understanding of the required data for learning will be needed. Directions for future work include developing better exploration methods specifically for offline training, and identification of experiences with high information for effective datasets.

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

| Domain | Task | $d_s, d_a$ | Description & Explore Suite Modifications |
|---|---|---|---|
| Ball-in-Cup | Catch
Catch Explore | 8,2 | Agent moves a cup to swing a ball attached via string into it. The explore suite version restricts the initial states to avoid trivial solutions. |
| Finger | Turn-Easy
Turn-Easy Explore
Turn-Hard
Turn-Hard Explore | 12,2 | Agent controls a finger, similar to a manipulator, to interact with objects. The explore suite version restricts the initial state of the finger to be away from the object. |
| Reacher | Turn-Easy
Turn-Easy Explore
Turn-Hard
Turn-Hard Explore | 6,2 | Agent controls a 2d manipulator to reach an arbitrary target state.
The explore suite version restricts the possible targets to a positive cone in front of the manipulator. |
| Walker | Stand
Stand Explore
Walk
Walk Explore | 24, 6 | Agent controls a 2d humanoid robot to move forward or stand. The explore suite version allows the robot to start on the ground rather than standing and has a sparse walking reward. |

Table 3: Extra information on the tasks used in the paper in DM Control Suite.

# A   APPENDIX

Here we include additional experimental context and results.

## A.1   ALGORITHMIC DETAILS

A summary of the exploration algorithm, Intrinsic Model Predictive Control is shown in Alg. 1. We utilize a distributed setup where multiple actors and learners can be deployed concurrently.

---

**Algorithm 1** Intrinsic MPC

---

**Given:** Randomly initialized proposal $\pi_\theta$, dynamics model $m_\phi$, reward model $r_i$, random critic $Q_\psi$. {*Modules to be learned*}
**Given:** Planning probability $p_{plan}$, replay buffer $\mathcal{B}$, MPO loss weight $\alpha$, learning rates & optimizers (ADAM) for the different modules. {*Known modules and parameters*}

{*Exploration loop – Asynchronously on the actors*}
**while** True **do**
    Initialize ENV and observe state $s_0$.
    **while** episode is not terminated **do**
        {*Choose between intrinsic planner and random action depending on $p_{plan}$*}
        {*Use learned proposal ($\pi_\theta$) as proposal for planner*}.
        sample $x \sim U[0,1]$
        $a_t \sim \begin{cases} \text{PLANNER}(s_t, \pi_\theta, m_\phi, r) & \text{if } x \le p_{plan} \\ \pi_\theta(s_t) & \text{otherwise.} \end{cases}$
        Step ENV$(s_t, a_t) \to (s_{t+1})$ and write transition to replay buffer $\mathcal{B}$
    **end while**
**end while**

{*Asynchronously on the learner*}
**while** True **do**
    Sample batch $B$ of trajectories, each of sequence length $T$ from the replay buffer $\mathcal{B}$
    Label rewards with reward model $r_i$.
    Update action-value function $Q_\psi$ based on $B$ using Retrace (Munos et al., 2016).
    Update model $m_\phi$ based on $B$ using multi-step.
    Update reward model $r_i$ based on $B$.
    Update proposal $\pi_\theta$ based on $B$ using (Byravan et al., 2021))
**end while**

---

The `PLANNER` subroutine takes in the current state $s_t$, the action proposal $\pi_\theta$, a dynamics model $m_\phi$ that predicts next state $s_{t+1}$ given current state $s_t$ and action $a_t$, the reward function model $r_i(s_t, a_t)$. Optionally, a learned state-value function $V_\psi(s)$ (with parameters $\psi$) that predicts the expected return from state $s$ can be provided. We use the Cross-Entropy Method (CEM) (Botev et al.), shown in Alg. 2.

---

**Algorithm 2** CEM planner

---

**Given:** state $s_0$, action proposal $\pi_\theta$, dynamics model $m_\phi$, reward model $r_i$, planning horizon $H$, number of samples $S$, elite fraction $E$, noise standard deviation $\sigma_{\text{init}}$, and number of iterations $I$.

{*Rollout proposal distribution using the model.*}
$(s_0, a_0, s_1, \ldots, s_H) \leftarrow \texttt{proposal}(m_\phi, \pi_\theta, H)$
$\mu \leftarrow [a_0, a_1, \ldots, a_H]$                                                    {*initial plan*}
$\sigma \leftarrow \sigma_{\text{init}}$
{*Evaluate candidate action sequences open loop according to the model and compute associated returns.*}
**for** $i = 1 \ldots I$ **do**
    **for** $k = 1 \ldots S$ **do**
        $p_k \sim \mathcal{N}(\mu, \sigma)$                                    {*Sample candidate actions.*}
        $r_k \leftarrow \texttt{evaluate\_actions}(m_\phi, p_k, H, r_i)$
    **end for**
    Rank candidate sequences by reward and retain top $E$ fraction.
    Compute mean $\mu_{\text{elite}}$ and per-dim standard deviation $\sigma_{\text{elite}}$ based on the retained elite sequences.
    $\mu \leftarrow (1 - \alpha_{\text{mean}})\mu + \alpha_{\text{mean}}\mu_{\text{elite}}$                  {*Update mean; $\alpha_{mean} = 0.9$*}
    $\sigma \leftarrow (1 - \alpha_{\text{std}})\sigma + \alpha_{\text{std}}\sigma_{\text{elite}}$         {*Update standard deviation; $\alpha_{std} = 0.5$*}
**end for**
**return** first action in $\mu$

---

## A.2   Additional Environment Details

The state-actions dimensions $(d_s, d_a)$ and descriptions for the environments used in this paper are detailed in Tab. 3. Additional collected reward distributions are shown in Fig. 11.

## A.3   Full Offline RL Agent-Task Performances

To supplement the results discussed in Sec. 4.3, we have included the performance per dataset size for all agents across all tasks. The results are shown in Fig. 13 and show the considerable variation when studying any given agent or task. There is substantially more variation across task than across agent, showing the value in continuing to fine-tune a set of tasks for benchmarking task-agnostic agents. The mean performance across all tasks is shown in Fig. 9, with a per-task breakdown shown in Table 4. A subset of agents and tasks are shown in Fig. 8 to show the convergence on a set of tasks.

## A.4   Additional Multi-task Learning Experiments

To compliment Sec. 4.4, we have included additional experiments for multi-task learning in the Reacher, Cheetah, and Walker environments, shown in Fig. 10. For these tasks, there is less clear of a benefit of using task-agnostic learning to generate data for offline RL policy generation. In our experience, this limited performance can be due to the fact that the environments are designed with specific behaviors and algorithms in mind, reducing the need for a diverse exploration method. A description of the starting state and the goal state for each task, as well as for the Pointmass and Finger environments from the main text, is available in Table 4.

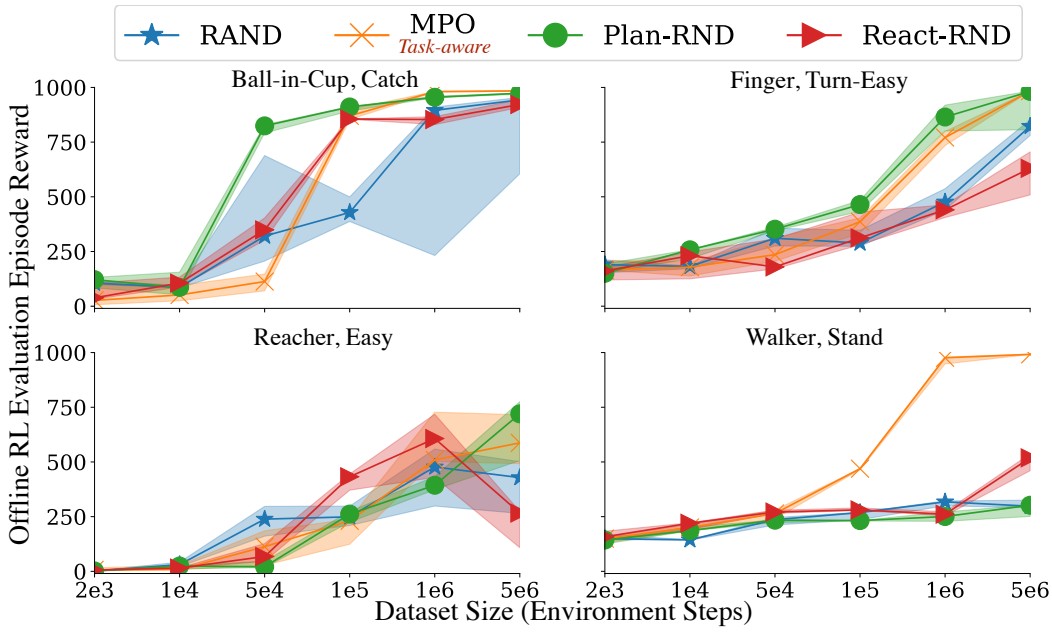

Figure 8: Highlighting performance of a subset of tasks and agents to showcase the relative differences that can emerge across tasks. The median, max and min across 3 training seeds are shown for one input dataset.

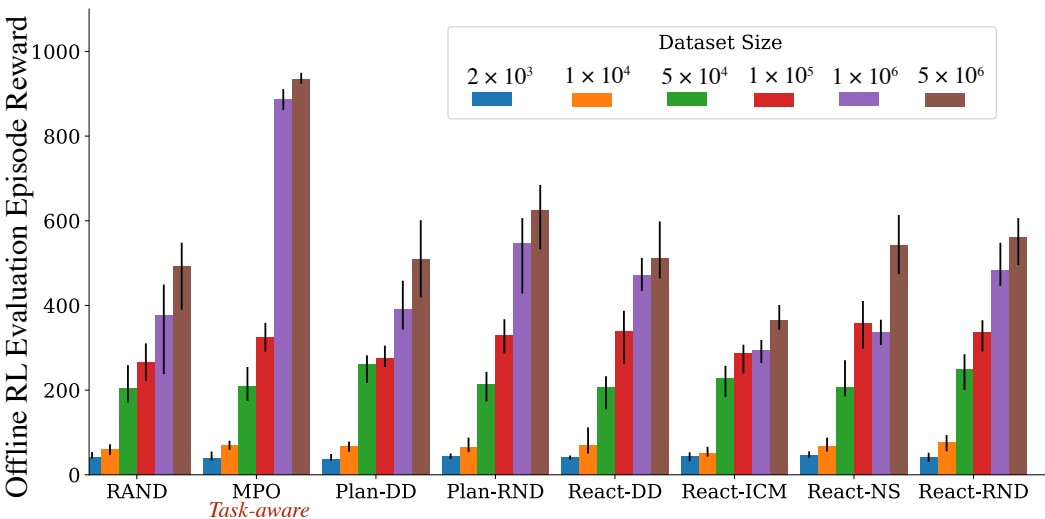

Figure 9: The median, max and min across 3 training seeds are shown for different dataset sizes. This figure shows how most of the exploration methods we evaluated performed similarly, yet our new IMPC agent with RND does have the best overall performance.

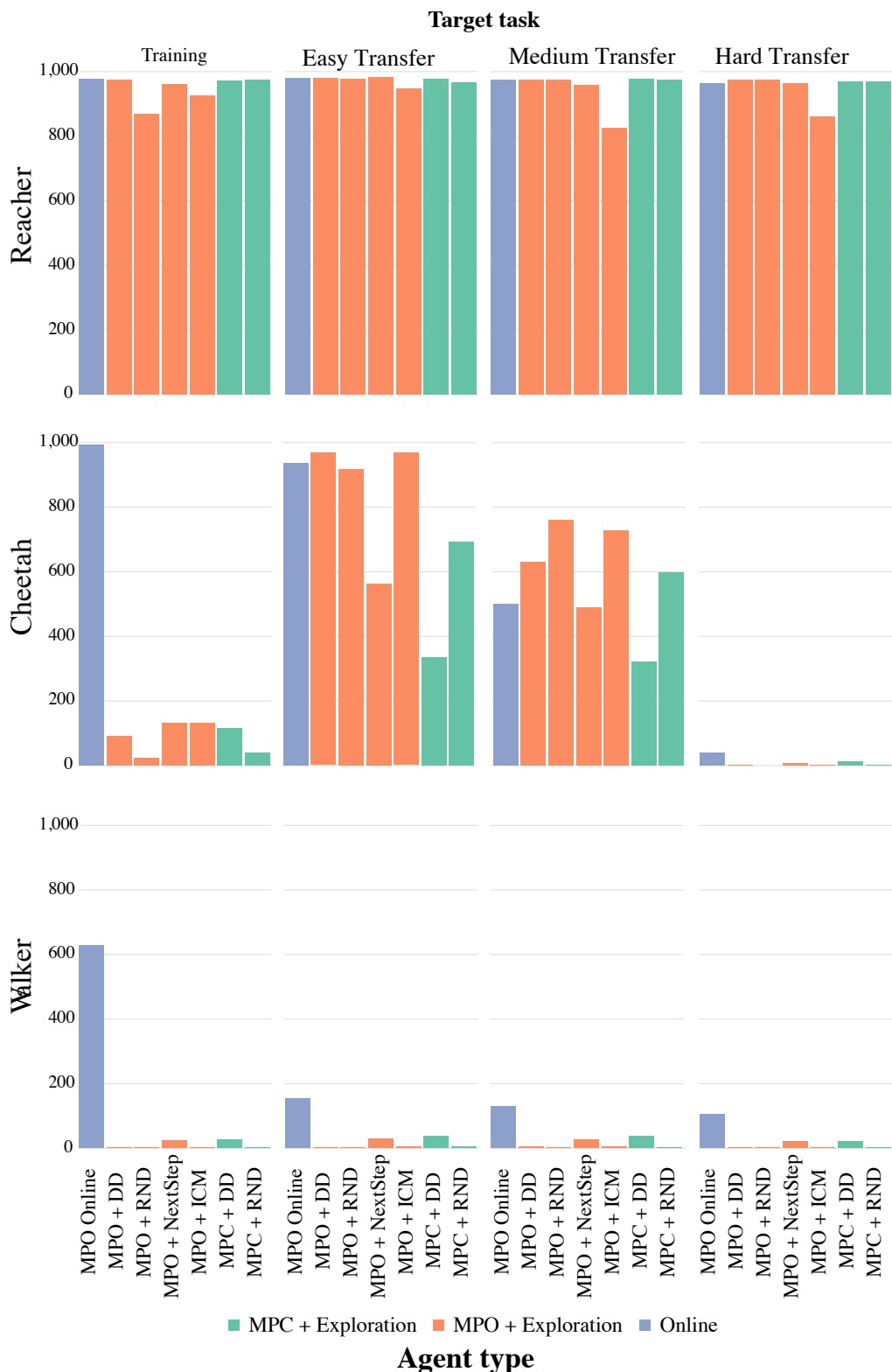

Figure 10: Performance across multiple tasks for offline RL agents trained on a single task-agnostic dataset.

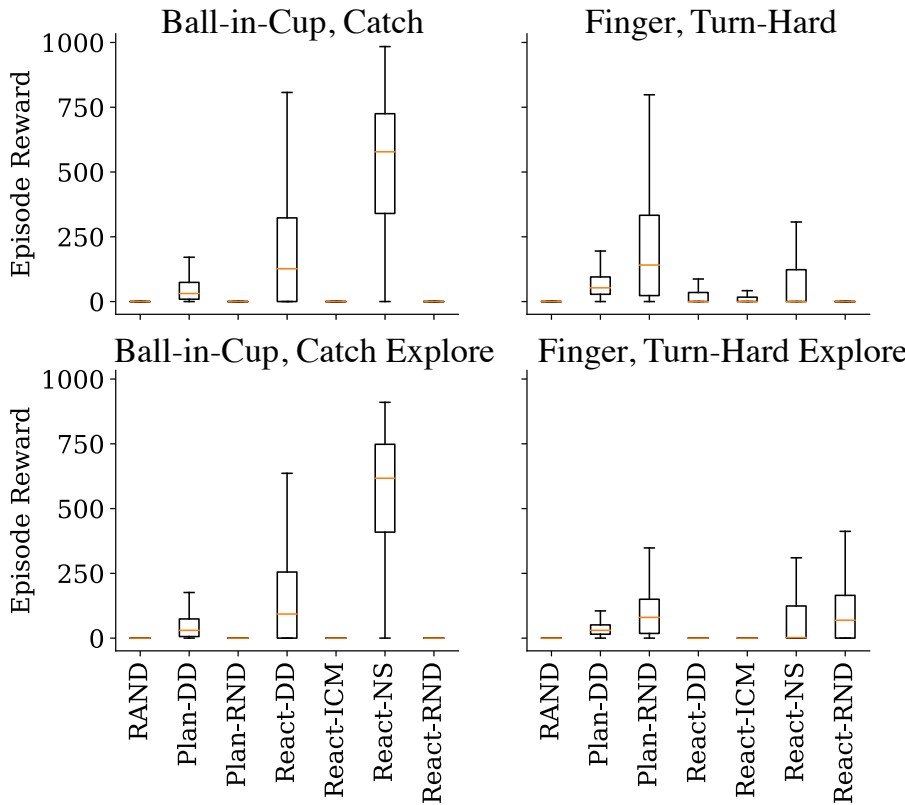

Figure 11: Collected reward per-episode (1000 environment steps) distributions across a subset of tasks and their Explore Suite variants. Boxplots depict the median episode reward (red line), 25th and 75th percentiles (box), and the maximum and minimum reward over 5000 episodes (whiskers). Across the suite of tasks we studied, there is often comparable variation among the exploration agents as the variation among tasks.

| Domain, Task | Task-Aware | Random | IMPC-DD | IMPC-RND | DD | ICM | NSM | RND |
|---|---|---|---|---|---|---|---|---|
| Ball-in-Cup, Catch | 984 | 943 | **973** | **973** | **975** | 901 | 972 | 922 |
| Ball-in-Cup, Catch Explore | 974 | 0 | 886 | 939 | **963** | 882 | 955 | 927 |
| Finger, Turn Easy | 983 | 822 | 768 | **982** | 577 | 541 | 689 | 630 |
| Finger, Turn Easy Explore | 969 | 916 | 636 | **944** | 850 | 170 | 920 | **948** |
| Finger, Turn hard | 977 | 544 | 486 | **975** | 513 | 366 | 510 | 567 |
| Finger, Turn hard Explore | 966 | 691 | 86 | 595 | 152 | 235 | **953** | **863** |
| Reacher, Easy | 767 | **955** | 881 | 901 | 626 | 612 | 550 | 868 |
| Reacher, Easy Explore | 974 | 613 | 254 | **741** | 98 | 89 | 235 | **814** |
| Reacher, Hard | 587 | 428 | 105 | **720** | **705** | 143 | 128 | 266 |
| Reacher, Hard Explore | 974 | 261 | 407 | 411 | **931** | 542 | **773** | 148 |
| Walker, Stand | 992 | 298 | **504** | 302 | 411 | 242 | 535 | **519** |
| Walker, Stand Explore | 983 | 244 | **259** | 81 | 188 | 232 | 179 | 256 |
| Walker, Walk | 978 | 147 | **446** | 97 | 103 | 60 | 107 | 84 |
| Walker, Walk Explore | 975 | 48 | **426** | 79 | 76 | 79 | 91 | 37 |

Table 4: The best performance with $5 \times 10^6$ transition datasets for all task-agnostic exploration agents and the task-aware MPO agent to compliment Table 2. To complement the table in the main text, bold represents the best reward among only task-agnostic agents across a task being within the range of $10^{th}$ and $90^{th}$ percentiles.

| Domain | Start state | Training | Easy Transfer | Medium Transfer | Hard Transfer |
|---|---|---|---|---|---|
| Pointmass $(x, y)$ | $(0, 0)$ | $(0.1, 0.1)$ | $(-0.1, -0.1)$ | $(0.3, 0.3)$ | $(0.2, 0.2)$ |
| Finger $(\theta)$ | $(0)$ | $(0.5)$ | $(1.0)$ | $(2.6)$ | $(-2.6)$ |
| Reacher $(x, y)$ | $(0.24, 0)$ | $(0, 0.1)$ | $(0, 0.2)$ | $(0, -0.1)$ | $(-0.1, 0.15)$ |
| Cheetah $(v)$ | $(0)$ | $(2.5)$ | $(-1.1)$ | $(1.1)$ | $(4.3)$ |
| Walker $(v)$ | $(0)$ | $(1.5)$ | $(0.5)$ | $(-0.5)$ | $(2.5)$ |

Table 5: Starting and goal states for each of the multitask environments. For Finger, $\theta$ represents the angle of a spinner which the finger can turn, and for Cheetah and Walker $v$ represents forward velocity.

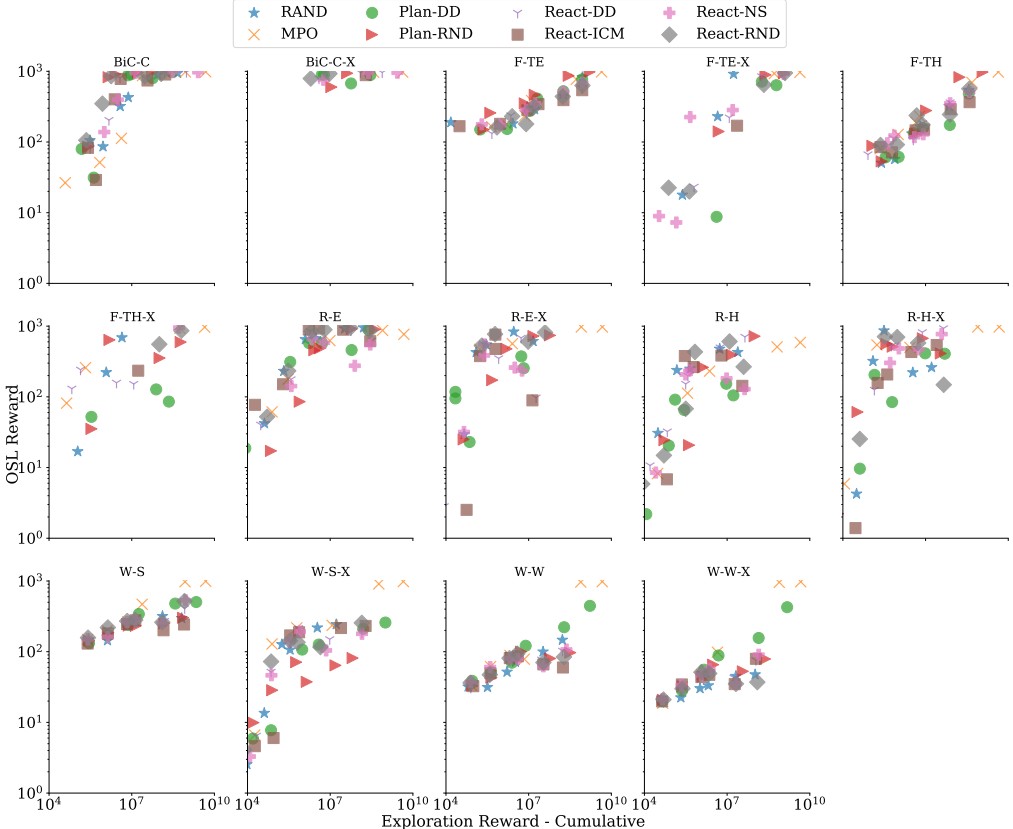

Figure 12: Correlation of final offline RL performance to cumulative reward in the training set. As the dataset sizes we use span many orders of magnitude of samples, both axes are plotted on a log-scale. Across all tasks, there is a trend of more reward in the training distribution relating to a better performing CRR agent.

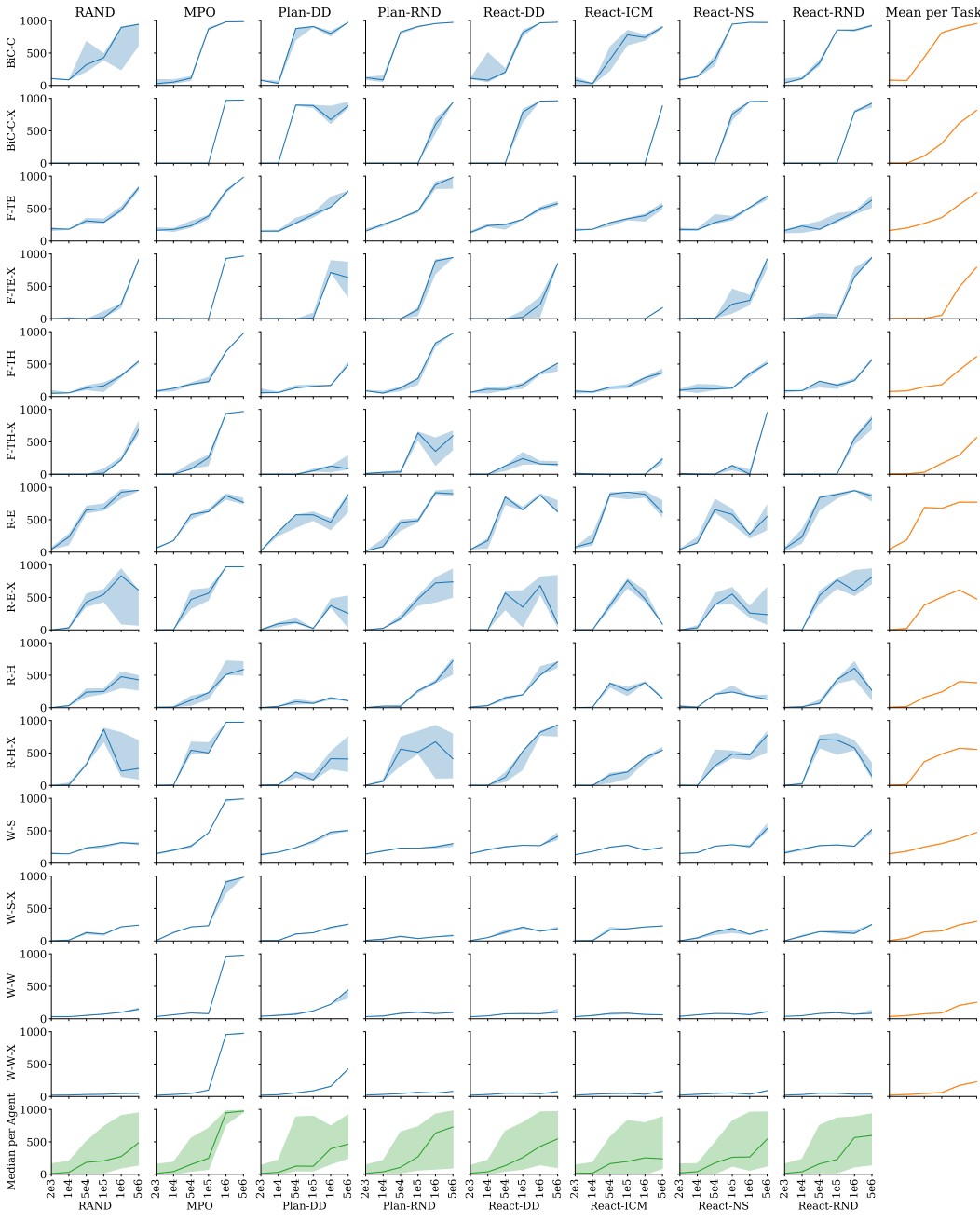

Figure 13: All agent-task pairings for final offline reinforcement learning performance. All tasks in the Explore and Control Suite occupy the rows while the Agents occupy the columns. The average results highlight the increased variation across tasks when compared to agents.

