# OpenReview forum: "The Challenges of Exploration for Offline Reinforcement Learning"
_ICLR.cc/2023/Conference — Submitted to ICLR 2023_

### Official Review · Reviewer_58zj · 2022-10-24

**Confidence:** 4
**Correctness:** 2
**Technical Novelty And Significance:** 2
**Empirical Novelty And Significance:** 3
**Recommendation:** 5

**Clarity, Quality, Novelty And Reproducibility:**

The paper is mostly clear, albeit some of the conclusions could be presented more directly.

The quality of the work could be improved (are the results only relevant for a single offline RL algorithm? Is dataset size really the importance variable?)

The work is relatively novel, despite it misses important discussion with respect to previous and parallel work in unsupervised exploration.

There seem to be no apparent reproducibility issue (3 seeds are not great, but I understand the possible computational constraints the authors might have).

**Strength And Weaknesses:**

Strengths:
- The problem addressed in the paper is of paramount importance: increasing our understanding of the kind of datasets that are needed to reach good performance with offline reinforcement learning has the potential to both shape the design of future offline RL algorithms and to have practical implications;
- The proposed MPC-based exploration method, which could be of independent interest, stands out as a clean way (probably cleaner than the reactive counterpart) of comparing the different exploration criteria.
- Despite being probably already known to some researchers in the field, having clearly written down some of the conclusions about the influence of dataset size and reward distribution could be useful to practitioners.

Weaknesses:
- The paper only uses the CRR offline RL algorithm for evaluating the downstream effects of the construction of different types of datasets. Despite this is a start, the actual outlook of offline RL algorithms used in the community is very vast, and there is really no baseline as established as for in the online RL case. This severely limits the scope of the work, since the reader cannot know a priori whether the conclusions of the study only hold for that specific offline RL algorithm or are indeed more general than that;
- The paper determines dataset size is possibly the most important variable to predict the performance for a downstream offline RL algorithm. However, this could be a bit misleading: one could imagine having the same trajectory over and over again in the dataset and clearly increasing the dataset size without any increase in the downstream performance. Despite this is a trivial counterexample, I think it is really epistemologically important to pin down the effect of diversity, more than size, on the resulting policy, because that one is more likely to be the underlying cause of the observed performance increase.
- The comparison with previous and parallel work is not very insightful from the perspective of the reader. For instance, it would be nice to understand more the relationship with (Yarats et al., 2022), instead of just briefly mentioning its existence. Moreover, there is some work similar in flavour about pretraining (https://arxiv.org/abs/2106.04799): it seems important to have a discussion on how the study presented in this work is different compared to that.

**Summary Of The Paper:**

The paper presents a study of a reinforcement learning setting consisting of a task-agnostic data collection phase and a task-aware offline optimization phases, on a set of continuous control tasks. Comparing the performance obtained when using different exploration criteria (combined with a new MPC-based method) in the data collection phase, as well as different dataset sizes, this work tries to draw conclusions about the features we should expect to find in offline RL datasets for a downstream algorithm to perform well.

**Summary Of The Review:**

Overall, I appreciate the direction presented in this work, as well as the general methodology that has been attempted. However, I believe there are some serious drawbacks that might overly reduce the scope of the conclusions drawn in the paper, and thus for now I tend to recommend rejection.

---

> ### Author Response · Authors · 2022-11-16
> **Reviewer 58zj Response**
>
> Thank you for your detailed review.
>
>
> We agree with the positive points you raised. Here we address the specific weaknesses you raise.
>
> * Reproducibility: For a brief comment on reproducibility, we agree that 3 seeds in any one aspect is not great, but when scaling across the number of domains and datasets we were using, it quickly became a lot. We collected 3 datasets in each environment for each exploration algorithm (7) (sometimes repeated across tasks), then for each dataset we needed to train multiple ORL policies. This also greatly impacted how hard it would be to use another ORL algorithm because we were already on 100s of runs that take multiple days on a single GPU.
> * Comparing ORL algorithms: As addressed above, we agree that there is always opportunity to improve papers by adding more experiments, such as with more algorithms. We believe that this paper is strengthened by the fact that independent work (different algorithms and implementation) got very similar results with a different focus of the paper (Yarats et al 2022 ExORL). The goal of this paper is increase the discussion around these important issues and set common practices for how this emerging subfield can be studied.
> * Data diversity: The reviewer makes an astute point on the diversity of data. It is well known in ML / RL that behavior cloning on exactly the same trajectory will not make a robust policy. Our paper discussing cumulative reward or multi-task settings definitely provides evaluation methods beyond dataset size. In fact, it was a little surprising how clear the trend of increasing data improving performance was (in fact, scaling relationships in RL have been less prevalent compared to other fields of ML).
> * Let us know if you find the explanation comparing our work to ExORL sufficient. We agree with you that it is very important and could be improved in the final version of the paper. This paper on re-training for downstream tasks is a great addition to the related work and we will add it.

---

### Official Review · Reviewer_BjG9 · 2022-10-24

**Confidence:** 3
**Clarity, Quality, Novelty And Reproducibility:** Please see my comments in the above s…
**Correctness:** 3
**Technical Novelty And Significance:** 3
**Empirical Novelty And Significance:** 2
**Recommendation:** 6

**Strength And Weaknesses:**

### Strengths
The paper is clearly written and easy to follow. The authors also conduct extensive experiments to investigate data collection in offline RL based on a number of intrinsic motivation based exploration algorithms to reveal interesting findings.

### Weakness
My main concern for the paper is its novelty. Although I acknowledge that the paper studies offline RL from a less-studied perspective (data collection instead of policy learning that aims to address extrapolation error) which is very interesting, it seems that it is very related to [1] without enough discussion about the differences. Specifically, [1] also studies data collection in offline RL, and it is worth discussing the differences between them.

[1] Denis Yarats, David Brandfonbrener, Hao Liu, Michael Laskin, Pieter Abbeel, Alessandro Lazaric, and Lerrel Pinto. Don’t change the algorithm, change the data: Exploratory data for offline reinforcement learning, 2022.

**Summary Of The Paper:**

The paper studies offline RL from a less-studied perspective - collecting informative experiences. The authors investigate the task-agnostic setting based on curiosity-based intrinsic motivation methods. The authors propose the Explore2Offline framework, and conduct an extensive empirical study to investing the effect of data collection strategies.

**Summary Of The Review:**

The paper studies offline RL from an interesting and less-studied perspective (data collection instead of policy learning), and propose a new exploration agent, Intrinsic Model Predictive Control that have strong performance. The authors also conduct extensive and in-depth analysis and investigation in standard offline RL tasks with different qualities. My main concern is its novelty, which is quite related to [1], but without enough discussion about their differences.

---

> ### Author Response · Authors · 2022-11-16
> **Reviewer BjG9 Response**
>
> Dear Reviewer, thank you for bringing up this shortcoming. We have added more commentary of the relationship between our two papers as we agree that is crucial to proper positioning of our paper. We have detailed the points above, but do not hesitate to respond if those are not clear!

---

### Official Review · Reviewer_QZYQ · 2022-10-25

**Confidence:** 3
**Clarity, Quality, Novelty And Reproducibility:** Code is not provided and detailed imp…
**Correctness:** 3
**Technical Novelty And Significance:** 3
**Empirical Novelty And Significance:** 2
**Recommendation:** 5

**Strength And Weaknesses:**

The idea and setting are novel and interesting. However, the method part is a little difficult to follow. the description of the IMPC is not clear.
1. What are the input and output of IMPC? How do you train it? What is the main contribution?
2. The experiments only choose MPO to verify the quality of the samples, which is not convincing enough. Since you propose a new framework and use the offline RL methods as a mechanism for evaluating exploration performance, I think more than 2 most popular methods should be used,
3. The paper does not give a deeper analysis of the data requirement of the offline RL. In other words, in order to get better performance, which kind of data is required by the offline RL methods? How to construct such a dataset?
4. There are some papers focusing on offline training and online fine-tuning to improve performance via fewer data. They also use exploration methods to guide online data collocation. Their methods can also be compared after slight modification.

**Summary Of The Paper:**

This paper explores how to collect informative data for offline RL methods. Many curiosity-based methods are considered to explore the environment. Intrinsic Model Predictive Control (IMPC) approach is proposed to improve the performance.

**Summary Of The Review:**

Please see my questions above.

---

> ### Author Response · Authors · 2022-11-16
> **Reviewer QZYQ  Response**
>
> Thank you for taking the time to improve the paper. We think the textual clarifications we have added will improve the impact of the paper.
>
> We have tried to clarify the IMPC introduction. If there is anything else you think is unclear, please let us know.
> 1. On IMPC:
>
> The input to IMPC is the current state of the agent and the output is the next action to be taken. The system uses two components, a learned dynamics model and a learned reward model. The models are both trained with standard supervised learning techniques on the replay buffer of agent experience.
>
> a. Regarding the contribution, this has been clarified in the text to expand on the original word “online” – the contribution is that the MPC formulation enables agents that are less prone to being impacted by stale data in the replay buffer.
>
> b. Specifically, the implementation is very closely inspired by this paper: https://arxiv.org/abs/2110.03363, which we better indicate in the paper.
>
> 2. I would like to clarify that CRR is the offline RL algorithm we used for generating policies and MPO is an online RL algorithm used for guiding the intrinsic agents to collect data. The focus of this paper was not to evaluate the differences between multiple off-policy or even offline RL algorithms, given their relative stability and accepted impact on the field, but rather to compare a large variety of dataset types and sizes.
>
> 3. Much like the recent work from Yarats et al., the paper shows that dataset size is particularly important to offline RL performance. Our paper takes this a step further than recent work and shows how correlated the actual reward in the dataset is to downstream policy performance. This is a great motivating factor for us to develop more sophisticated data measurement tools to RL in the future, where automated comparison across agent trajectories is accepted to be a challenging task.
>
> 4. Thanks for pointing this out. We have added a citation to AWAC (https://arxiv.org/abs/2006.09359) which discusses this directly. Are there any other papers you had in mind?

---

### Author Response · Authors · 2022-11-16
**General Response to Reviewers**

Dear Reviewers,

Thank you for your time spent on improving this paper. Here will address some of the primary discussion points around the paper and steps made to improve them. We have updated the paper accordingly. Following, we will reply to each reviewer directly for finer points.

## Explore2Offline and ExORL

As multiple reviewers mentioned, our work is closely related to recent work from Yarats et al. 2022 (ExORL). The works are very similar, but implemented independently.

The contributions of our paper relative to theirs can be summarized as follows:
Explore2Offline showcases the importance of data collection in vanilla environments and those designed to be more challenging exploration problems (Explore Suite). ExORL focuses more on evaluating existing algorithms on known environments, and is extremely thorough in doing so.
Explore2Offline also discusses how this data could be more directly collected, through the use of a method like IMPC that suffers less from reward-staleness.
Explore2Offline also compares to an existing online RL algorithm, MPO, which shows the gap that intrinsic methods need to close to match performance in the task-aware setting.

The related works section has been updated to reflect this, and their paper is also mentioned in the introduction of our paper.

## Comparison across ORL algorithms

Multiple reviewers agree that the impact of our work would be *increased* if we were to compare more offline RL algorithms. As the reviewer correctly mentions and the field matures, there are many existing and functioning offline RL algorithms. While the direct implementation of CRR we used is not ready for release at this time, the algorithm has also been used successfully by multiple other projects. CRR closely relates to many other algorithms in the literature, with details being found in the original paper (see here https://proceedings.neurips.cc/paper/2020/file/588cb956d6bbe67078f29f8de420a13d-Paper.pdf) – specifically these comparisons are made in the second half of 2. Related work and the concluding paragraphs in 3.2 Policy Learning with CRR.

The goal of the paper is to provide commentary on the impact of the data, independent of an algorithm. In order to maximize the amount of compute and analysis done on this question, repeating over many algorithms becomes challenging (this paper already used a rather large amount of compute).

## Other points

In addition to these points, we are addressing a few other comments from the reviewers, including:
Clarifying the implementation of IMPC,
Discussion of the take-aways of the paper on what specific properties data should have,
Discussion of other offline data-efficient methods,
Discussion of data diversity in addition to data size,
And more minor clarifications.

Thank you!

---

### Decision · Program_Chairs · 2023-01-20

**Decision:**

Reject

**Justification For Why Not Higher Score:**

The paper studies an interesting problem about how to collect data so that it can useful for downstream offline RL tasks. Yet, several reviewers are concerning about the novelty as existing works may have studied similar approaches and settings. The most concerned part is the lack of comparison with other offline algorithms, which may produce different effects on the data size.

**Justification For Why Not Lower Score:**

N/A

**Metareview: Summary, Strengths And Weaknesses:**

Summary:
The paper studies how to coleect data for offline RL. The main approach consists of a task-agnostic data collection phase and a task-aware offline optimization phases, on a set of continuous control tasks. The paper proposed an algorithm called Explore2Offline based on this idea and conduct an extensive empirical study to investing the effect of data collection strategies.

Strength:
- The paper is clearly written and easy to follow. The authors also conduct extensive experiments to investigate data collection in offline RL based on a number of intrinsic motivation based exploration algorithms to reveal interesting findings.
- Despite being probably already known to some researchers in the field, having clearly written down some of the conclusions about the influence of dataset size and reward distribution could be useful to practitioners.

Weakness:
- Missing related works about reward-free RL with function approximations
- Reviewers are concerned about the presence of a single offline RL algorithm in all of the experiments presented in the paper. It might be necessary to consider different offline RL algorithms
- The comparison with previous and parallel work is not very insightful from the perspective of the reader.